# An Implementation Science Framework to Develop a Clinical Decision Support Tool for Familial Hypercholesterolemia

**DOI:** 10.3390/jpm10030067

**Published:** 2020-07-23

**Authors:** Hana Bangash, Laurie Pencille, Justin H. Gundelach, Ahmed Makkawy, Joseph Sutton, Lenae Makkawy, Ozan Dikilitas, Stephen Kopecky, Robert Freimuth, Pedro J. Caraballo, Iftikhar J. Kullo

**Affiliations:** 1Department of Cardiovascular Medicine, Mayo Clinic, Rochester, MN 55905, USA; Bangash.hana@mayo.edu (H.B.); Gundelach.justin@mayo.edu (J.H.G.); Dikilitas.ozan@mayo.edu (O.D.); kopecky.stephen@mayo.edu (S.K.); 2Center for Science of HealthCare Delivery, Mayo Clinic, Rochester, MN 55905, USA; Pencille.laurie@mayo.edu; 3User Experience Research, Saharafox Creative Agency, Rochester, MN 55905, USA; amakkawy@saharafox.com (A.M.); lenaebeth@gmail.com (L.M.); 4Department of Information Technology, Mayo Clinic, Rochester, MN 55905, USA; Sutton.joseph@mayo.edu; 5Department of Digital Health Sciences, Mayo Clinic, Rochester, MN 55905, USA; Freimuth.robert@mayo.edu; 6Department of General Internal Medicine, Mayo Clinic, Rochester, MN 55905, USA; Caraballo.pedro@mayo.edu

**Keywords:** electronic health record, clinical decision support, CDS, familial hypercholesterolemia, FH, genomics

## Abstract

Electronic health record (EHR)-based clinical decision support (CDS) can address the low awareness and undertreatment of familial hypercholesterolemia (FH), a disorder associated with a markedly increased risk of coronary heart disease. We aimed to incorporate provider perspectives into the development and implementation of a CDS tool for FH. An implementation science framework and a user-centered design process were used to create a CDS tool for FH. Primary care physicians and specialist physicians participated in qualitative interviews, usability testing and an implementation survey. The CDS was configured in two formats—a best practice alert (BPA) and an in-basket message and subsequently deployed in the EHR in silent mode. The key themes that emerged from the analysis of interview transcripts included understanding and awareness of FH, clinical workflow, physician preferences and value of CDS tools, perspectives on patient needs and values and dissemination and implementation. Recommendations related to usability included preferred CDS format and placement, content, timing and frequency, and level of alert urgency/prioritization. In response to the survey, 84.6% of physicians agreed that the CDS would improve early FH diagnosis and 92.3% agreed that it would help them identify and manage FH patients. Physician feedback led to iterative CDS refinement. In summary, we developed a CDS tool for FH using an implementation science framework and physician feedback. Initial deployment revealed a significant burden of FH and the potential for the CDS tool to have a large impact.

## 1. Introduction

Familial hypercholesterolemia (FH), is often monogenic in etiology, remains vastly undiagnosed and untreated in the United States and is therefore an ideal use case for the development of a clinical decision support (CDS) tool aimed at improving case detection and treatment as well as promoting cascade testing [1,2,3]. Awareness of FH is low among both patients and providers [4,5]. Providers in both primary care as well as specialty clinics such as cardiology often perceive all cases of hypercholesterolemia similarly, failing to recognize the higher risk of coronary heart disease (CHD) and the need for familial testing when the etiology is genetic. Initiation of appropriate lipid lowering medications is often delayed and goal low-density lipoprotein cholesterol (LDL-C) levels are achieved only in a minority of FH patients [1,6].

In the face of the ever-increasing volume and complexity of medical data, the delivery of CDS to health care providers at the point-of-care is an urgent need. Nowhere is this need more acute than for the practice of genomic medicine in the primary care setting where providers are often uncertain about the interpretation and application of genomic test results to patient care [7,8]. In such settings, CDS tools can provide guidance related to the interpretation of genomic test results and their subsequent application to patient management [9,10]. However, healthcare providers often express concern about alert fatigue and poor usability/functionality of CDS tools, which may partly stem from poor design and a lack of participation by end users during the development phase [11,12,13]. Disappointingly, electronic health record (EHR) systems have added complexity to clinical workflows, reduced provider–patient interactions and increased cognitive burden and burnout [14].

A potential means of making EHR systems more user friendly and ‘intelligent’ is by integrating CDS tools that are designed and implemented based on input from relevant stakeholders such as providers. This could, in turn, reduce provider cognitive burden and increase efficiency and satisfaction [15,16]. Implementation science, defined as the study of methods to facilitate uptake and integration of evidence-based health interventions into routine practice, can be applied to better understand the barriers and facilitators of CDS implementation. The use of an implementation science framework can ensure that CDS tools are optimally designed and integrated in clinical practice to enhance point-of-care management [17,18,19,20,21,22]. We therefore used an implementation science framework to develop a CDS tool for FH based on physician feedback from qualitative interviews, usability testing and an implementation survey.

## 2. Materials and Methods

This study was considered exempt by the Mayo Clinic Institutional Review Board and was conducted between November 2018 and October 2019.

### 2.1. Participants and Recruitment

The target sample size for the qualitative interviews and usability testing was ~8–10 participants based on evidence from usability studies, which indicates that up to 80–85% of usability issues can be identified by the first eight participants [23,24]. To achieve the target sample size, we conducted purposive sampling with the following inclusion criteria: (i) staff physicians at the Mayo Clinic campus in Rochester, Minnesota; (ii) from primary care including the departments of family medicine and community internal medicine as well as specialty fields such as cardiology and vascular medicine; and (iii) with previous experience using institutional CDS tools. Physicians were contacted by templated emails that contained invitations to participate in the interviews and to avoid potential coercion; they were not contacted by a supervisor.

### 2.2. Physician Interviews and Usability Testing

Interviews with physicians were conducted from November 2018 to February 2019. Qualitative methodologies and a user-centered design process were applied within a type 1 effectiveness-implementation hybrid framework [24,25,26,27,28,29]. Semi-structured qualitative interviews and usability testing sessions were conducted to obtain physician perspectives on two EHR-based CDS prototypes. The first prototype was a best practice alert (BPA)—A clinical reminder with evidence-based management guidelines that would passively display to providers (requiring them to click on the alert to open it) at the point-of-care. The second prototype was an asynchronous in-basket alert—A type of inbox message that contained patient laboratory data and would be visible to providers upon logging into the EHR. Both CDS formats (BPA and in-basket) were linked to an automated detection algorithm in the EHR that identified individuals as ‘possible FH’ cases, defined as LDL-C ≥ 190 mg/dL in the absence of secondary causes of hypercholesterolemia. Both CDS formats were presented during each interview and the “Think Aloud” usability technique was applied; physicians were asked to talk out loud and verbalize their thoughts while interacting with the CDS prototypes [30,31,32]. Four rounds of physician interviews and usability testing were conducted—new physicians, without prior exposure to the CDS tool, participated in each round and their feedback informed CDS prototyping and iterative refinements in subsequent rounds.

Interviews and usability testing were conducted by research study team members, including a physician, a qualitative researcher, a user experience/user interface expert and a research program coordinator. Each interview lasted one hour, was audio recorded and transcribed. Participants were given lunch during the interview and received a USD 100 honorarium. A semi-structured interview guide was developed and tested in a ‘mock’ interview to ensure that the questions were comprehensive (see Appendix A). The results from the mock interview were not included in the thematic analysis.

### 2.3. Implementation Survey

Following each interview, physicians were asked to complete a twenty five-question implementation survey to elicit multi-level contextual factors that could influence CDS implementation in clinical practice (see Appendix A). Modified from Weiner et al., twelve survey questions assessed three implementation outcome measures: Acceptability of Intervention Measure (AIM), Intervention Appropriateness Measure (IAM) and Feasibility of Intervention Measure (FAM) [33]. The remaining thirteen survey questions were modified from the Consolidated Framework for Implementation Research (CFIR) [34,35].

### 2.4. Deployment of CDS in Silent Mode

Both the BPA and in-basket CDS formats were deployed in ‘silent mode’ across all Mayo Clinic sites, including the Mayo Clinic Health System (MCHS) for a three-month period (July 2019 to October 2019). The silent mode setting would trigger the CDS alert in the EHR for providers in primary care, general internal medicine and cardiovascular medicine; however the alert was set to not display for providers, triggering only in the EHR background. The goal of deploying in silent mode was to gather initial metrics on the triggering of CDS at different sites and use these metrics as a measure of the burden of ‘possible FH’. Each alert was cued by a different action; the BPA fired when a patient’s EHR was opened, while the in-basket fired when results of a newly ordered lipid were placed in the EHR.

### 2.5. Data Analysis

The Framework Method was applied to transcript coding [36,37]. An inductive approach was used to generate themes from the transcribed data through open coding of transcripts. An initial codebook was developed by two qualitative researchers (HB and LP) who independently coded two transcripts line-by-line and then read each transcript in a group setting to reach a consensus on applied codes and ensure inter-rater reliability. After the codebook was established, the remaining transcripts were coded independently by two research team members and any inconsistencies in assigned codes were resolved through group discussion at bi-weekly meetings. All transcribed data were imported into ‘Reframer’—A qualitative research software tool (Optimal Workshop 2015, Wellington, New Zealand)—And each code was built into the tool as a ‘tag’ that could be applied to multiple observations. Survey responses of primary care physicians (PCPs) and specialists were compared using Wilcoxon rank-sum test to identify the median score, interquartile range (IQR) and any statistically significant (*p* < 0.05) differences in the responses between PCPs and specialists. Descriptive data, including frequency of responses, were also noted.

## 3. Results

Thirteen physicians from primary care and cardiovascular medicine were recruited to the study and gave informed consent prior to participation in the qualitative interviews and usability testing—seven were primary care physicians (PCPs) and six were specialist physicians. Participants included physicians in leadership roles and those serving on institutional committees. Participant demographic characteristics are summarized in Table 1.

### 3.1. Physician Interview Findings

Analysis of the physician interview transcripts identified five key themes and eleven subthemes along with representative quotes (Table 2).

#### 3.1.1. Understanding and Awareness of FH

This theme encompassed physicians’ current awareness and knowledge of FH and their perceived role in the detection and management of FH patients.

1.Subtheme: Physicians awareness of FHAmongst both PCPs and specialists, there was a lack of FH awareness—physicians did not always distinguish between hypercholesterolemia of a genetic etiology versus hypercholesterolemia due to other causes. In some instances, both were perceived as being similar and requiring similar management.

2.Subtheme: Physicians perceived scope of workPCPs and specialists agreed that all physicians were responsible for diagnosing FH; however, they differed in their views on who was responsible for the management of FH patients. Some physicians preferred to manage FH patients themselves, while others preferred referral to a specialist.

#### 3.1.2. Clinical Workflow

Clinical workflow aimed to understand physician workflows in primary care and specialty settings and gain insights into the next steps physicians were likely to take after a patient had been diagnosed with FH.

1.Subtheme: Diagnosis in workflowDetermining how physicians established a diagnosis in workflow centered on understanding when tasks such as ordering laboratory tests and reviewing results were performed. Physicians identified CDS alert relevance as being based on both encounter type and patient type. PCPs with high patient volumes were likely to have limited time to review alerts.

2.Subtheme: Next steps for physiciansFor a patient with suspected FH, the next steps for physicians included prescribing medication, counseling on lifestyle changes, ordering tests and ruling out secondary causes of hypercholesterolemia. Most physicians were willing to refer patients to the institutional FH Clinic if this was the appropriate next step in management.

#### 3.1.3. Physician Preferences and Value of CDS Tools

This theme encompassed physician perspectives on CDS tools and whether these tools impacted provider cognitive burden and time. Physicians also gave insights into their perceived value of the institutional FH Clinic.

1.Subtheme: CDS tool preferencesMost physicians had previously used one or more CDS tools and viewed them positively. However, a physician’s personal style or prior knowledge of a patient was likely to influence their decision on whether or not to use a specific CDS tool. With a relatively recent transition to a new EHR system at the time of interviews, some physicians continued to face challenges in interacting with the new EHR and highlighted the need to find more efficient ways to carry out routine tasks.

2.Subtheme: Cognitive loadA common feedback from all physicians was that the increasing number of CDS alerts and in-basket messages were contributing to rising levels of alert fatigue and information overload, subsequently resulting in CDS alerts being ignored or bypassed.

3.Subtheme: Value of FH Clinic referral to physiciansPhysicians agreed that having the option to refer patients with ‘possible FH’ to the institutional FH Clinic was likely going to reduce their cognitive burden and would be useful for complex cases such as those where: (a) goal LDL-cholesterol was not being attained on maximum statin therapy; (b) a patient had statin intolerance or (c) cases where genetic testing was warranted.

#### 3.1.4. Perspectives on Patient Needs and Values

Physicians highlighted the importance of engaging with patients through digital tools and also shared their insights on the value of the institutional FH Clinic to both patients and their at-risk family members.

1.Subtheme: Patient engagement through the use of digital toolsPhysicians indicated that there was a need for patient education materials that could be printed and attached to messages as well as videos that could be shared directly through the institutional patient portal. Most physicians agreed that there was value in shared decision-making tools such as patient decision aids, especially if these were available on the internet and easily accessible. Physicians also highlighted the importance of being able to share the computer screen displaying the decision aid with patients, so as to engage them during clinical encounters.

2.Subtheme: Value of FH Clinic referral to patientsPhysicians highlighted that the value of a referral to the institutional FH Clinic for patients and their family members centered on: (a) obtaining access to resources available specifically in the FH Clinic; (b) cascade testing of family members; (c) receiving access to new medications such as PCSK9 inhibitors and (d) facilitating appointment scheduling for genetic testing.

#### 3.1.5. Dissemination and Implementation

This theme encompassed multi-level barriers and facilitators that could potentially impact the adoption of CDS in clinical practice.

1.Subtheme: Facilitators to Dissemination and ImplementationPhysicians suggested that departmental meetings, conferences, grand rounds as well as newsletters could be used to promote awareness of the CDS tool and facilitate its uptake in practice. They also highlighted the need to obtain support from institutional champions, opinion leaders, early adopters and others likely to influence the attitudes and behaviors of providers.

2.Subtheme: Barriers to Dissemination and ImplementationThe main barriers identified by physicians that were likely going to impact utilization of the CDS tool in practice included use of pre-existing CDS tools that may prevent adoption of new tools, limited time in patient encounters to view and act upon CDS alerts and lack of institutional guidance and education on new CDS availability.

### 3.2. Usability Recommendations

The usability testing of both CDS prototypes led to physician feedback that broadly fell into one of four recommendation categories summarized in Table 3.

### 3.3. Implementation Survey Results

The majority of PCPs and specialists responded positively to questions that assessed implementation outcome measures; 84.6% (*n* = 11) of physicians indicated that they liked the tool and it seemed easy to use (Table 4). Similarly, positive responses were noted for questions pertaining to the CFIR constructs; 84.6% (*n* = 11) of physicians indicated that the CDS tool would improve early diagnosis of patients with FH, while 92.3% (*n* = 12) of physicians agreed that the CDS tool would help them identify, refer or manage FH patients. However, in response to the statement ‘this tool will not increase the time needed with a patient’, 69.2% (*n* = 9) of physicians were either neutral or negative in their response. Overall, there was agreement in responses between both PCPs and specialists (see Appendix A).

### 3.4. Silent Mode Metrics

The rapid prototyping of the BPA and in-basket continued until no new physician feedback was obtained from three consecutive interviews and thematic saturation was attained. The final BPA and in-basket were then deployed in the EHR in ‘silent mode’ for gathering initial metrics (Figure 1). During a three-month period, the BPA triggered 5415 times for 1440 unique patients, while the in-basket triggered 953 times for 924 unique patients at the three major Mayo Clinic sites and the MCHS.

## 4. Discussion

The dearth of usable and relevant digital tools is a major hurdle to the implementation of genomic medicine [7,38]. In this report, we describe the creation of a genomic CDS tool for FH, using an implementation science framework aimed at evaluating CDS effectiveness and gaining early insights into clinical implementation through physician feedback. The relatively high prevalence but yet low awareness and detection of FH motivated the development of a CDS tool for FH. The work described herein could inform the creation of CDS tools for other genomic disorders in diverse settings.

The introduction of EHRs has been associated with increased provider cognitive load, reduced professional satisfaction and increased rates of burnout [39,40]. To address these concerns, the Office of the National Coordinator for Health Information Technology (HIT) proposed aligning EHRs and tools with clinical workflows, improving user interfaces to increase efficiency and implementing health IT based on stakeholder engagement [41]. This includes the creation of well-designed, usable CDS tools that can fill gaps in knowledge and provide guideline-based recommendations at the point-of-care.

We conducted qualitative interviews and usability testing with both PCPs and specialists; PCPs are often the first line of contact for most patients in the community setting, especially for needs such as the evaluation of lipids. They are also more likely to treat multiple family members and thus can play a pivotal role in increasing FH detection and screening within a community. However, physicians identified a number of barriers that were likely to reduce CDS utilization in clinical practice—the most common barrier being the increasing cognitive burden on providers due to EHR complexity and limited time during clinical encounters. These findings are supported by previous studies that examined barriers to CDS implementation [9,42,43].

To promote CDS adoption in clinical practice, physician feedback emphasized the need to deploy passive alerts that would be less disruptive to workflows. They also highlighted that in-basket message alerts should be linked to new lipid panel reports to simplify workflows. Physicians recommended shortening and reorganizing the alert content for clarity and embedding a link to an order set specific for FH so as to reduce the number of clicks. Physicians also recommended including a link to AskMayoExpert, a Mayo Clinic knowledge resource with a topic module dedicated to FH that provides information on clinical presentation, diagnostic criteria and management [44]. AskMayoExpert also has links to additional knowledge modules, thereby enabling providers to access a number of relevant topics without workflow disruption [45].

Another important insight was that concurrent to the development of a CDS tool, an attempt should be made to increase awareness of the tool, provide education on how to use it and obtain support from institutional leadership. This feedback led to efforts to increase FH awareness amongst providers and highlight the CDS tool for FH at the Mayo Clinic Rochester campus. Rapid review pocket cards, which gave an overview of FH, were developed and distributed to PCPs and specialists. Several grand rounds were conducted across the institution by FH physician experts to disseminate knowledge and highlight the available CDS tool.

The implementation survey provided physician perspectives on contextual variables that were likely to influence CDS adoption in clinical practice. Most physicians indicated a receptive attitude towards CDS integration. The only survey item where the majority of the physicians gave either a neutral response or disagreed was regarding the CDS tool ‘not’ increasing time spent with a patient. This response reiterated the need for CDS to be designed to increase efficiency and not add to provider burden.

A relatively large number of both BPA and in-basket alerts fired during silent-mode deployment, indicating a high burden of ‘possible FH’. This suggests a potentially large impact of CDS deployment on health outcomes. Currently, both alerts remain deployed in silent mode and will be further evaluated in a clinical trial to assess the effect of the CDS on outcomes such as changes in patient management post-deployment.

A limitation of our study is that interviews were limited to physicians; the input of nurse practitioners, physician assistants, and residents in training may have relevance for clinical implementation of CDS tools in differing clinical workflows. Additionally, inherent EHR limitations prevented some physician feedback from being implemented in the final CDS tools that were deployed in the EHR. The number of physicians recruited for the study was modest; however, evidence from usability studies indicates that such a number is adequate for developing and establishing the tool and to initiate the application of the tool in clinical practice [23,24]. After initial implementation of the CDS tool, metrics for its use, including the utilization of the FH order set and AskMayoExpert knowledge resource will be evaluated to determine need for refinements and modifications. Such modifications, based on the pilot implementation, could further improve the FH CDS tool for use at the point-of-care.

## 5. Conclusions

Employing an implementation science framework with qualitative interviews and usability testing, physician feedback was obtained to guide the development of a CDS tool for FH and integrate it with the EHR for providers in primary care and cardiovascular medicine. A relatively large number of alerts fired during CDS deployment in silent mode, indicating a high burden of ‘possible FH’ and the potential for CDS to increase the point-of-care detection of FH. The process described here, along with the physician feedback obtained, can inform the creation of CDS tools for genomic disorders at other institutions and health systems.

## Figures and Tables

**Figure 1 jpm-10-00067-f001:**
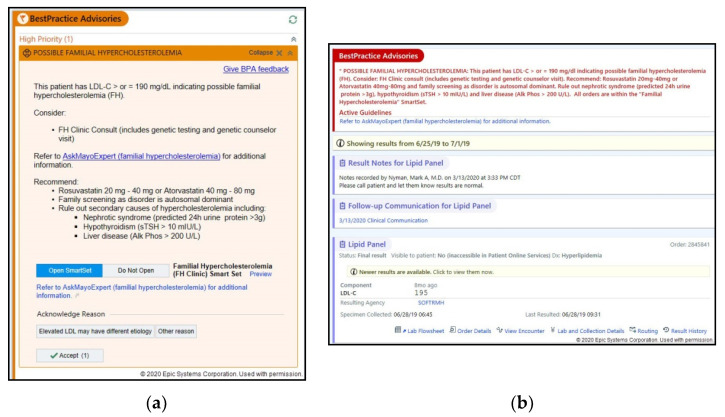
(**a**) The final best practice alert or BPA deployed in the EHR. (**b**) The final in-basket message linked to a lipid panel report, deployed in the EHR in silent mode.

**Table 1 jpm-10-00067-t001:** Physician characteristics (*n* = 13).

Physician Characteristics	*n* (%)
Gender	
Females	7 (53.8)
Males	6 (46.2)
Age	
<40 years	4 (30.8)
40–60 years	7 (53.8)
>60 years	2 (15.4)
Race/Ethnicity	
Non-Hispanic white	10 (76.9)
Black	1 (7.7)
Asian	1 (7.7)
Hispanic	1 (7.7)
Specialties	
Community Internal Medicine	3 (23.1)
Family Medicine	3 (23.1)
Family Medicine/Obstetrics	1 (7.7)
Cardiology	5 (38.4)
Vascular Medicine	1 (7.7)
Years in Practice	
0–5	3 (23.1)
6–10	1 (7.7)
11–15	2 (15.4)
16–20	1 (7.7)
More than 20	6 (46.1)

**Table 2 jpm-10-00067-t002:** Themes and representative quotations identified from physician interviews.

Theme	Quotation
Understanding & Awareness of FH	“But the initial management as far as I know is the same as anybody else.” (PCP)
	“…I mean, I see people that carry that diagnosis [FH], but I don’t think I would, sort of, come up with it myself.” (Specialist)
Clinical Workflow	“And to be honest, if I’m seeing them for strep throat, I might ignore it [CDS] or mention to them that they need to talk to their doctor about it. I will try to mention to a patient ‘hey you’re due for your lipids, you’re due for your colonoscopy, may I order those things for you?’ and if they say ‘yes’, I will, but if I have a 15-min appointment for their broken leg… I don’t know that I’m going to get into a huge discussion…” (PCP)
	“I would counsel them, and then I would want to send them to the FH Clinic if they’re around or local. If not, you know, of course I’d start them on a statin, and then you’re kind of stuck because I wouldn’t feel comfortable ordering the genetic tests for them. That’s really the logical next step, in my opinion. In which case, you’d try to send them to Genetics and/or the FH Clinic.” (Specialist)
Physician Preferences & Value of CDS Tools	“I would suggest one area that we constantly struggle with is finding the right orders in Epic when it’s orders that we haven’t done a lot of… so the more detail… showing you exactly how to find the right order would be very helpful… Because we have really struggled… with the amount of different things that we order, that has been a real hard spot…” (PCP)
	“Personally, I guess that’s why I’m here, I would really prefer passive alerts… I think, where it was an active alert… people get alert fatigue, and they’re just going to click it to bypass it.” (PCP)
Perspectives on Patient Needs & Values	“Because I’ve learned when I do risk counseling and I use the shared decision-making aid, which includes all the risk percentages…I’m learning as I teach my own patients, like, oh, yeah, that’s right, smoking does increase by this much, and so if you have something like that for FH, if you really want to drive home the point of how much greater risk, people who use that will start educating themselves in addition to their patients.” (Specialist)
Dissemination & Implementation	“I think whenever anything significantly new like this [CDS] is deployed; some type of communication is useful. I mean either in the EHR update, which many of us are actually reading now, or in like communication from leadership or through multiple approaches.” (PCP)
	“The problem with BPA is, again, that just so many of them don’t apply. You really have to sort it out from them. You’d have to suppress everything that didn’t apply to me and patients…I’m just going to see a bunch of yellow and I’m going to ignore it because it’s not… it doesn’t apply to me, I’m therefore going to ignore everything and I’m going to miss the important alarms.” (Specialist)

Abbreviation: PCP, Primary care physician; FH, Familial hypercholesterolemia; CDS, Clinical decision support.

**Table 3 jpm-10-00067-t003:** Usability recommendations with corresponding descriptions and representative quotations highlighted from usability testing with providers.

Recommendation	Description	Quotation
Format & Placement	Having both the BPA and in-basket alert formats embedded in the EHR would likely increase the probability of providers viewing the CDS	“And if you could somehow attach to the results in-basket that would allow us to see that alert at the same time that we’re seeing the result, which would be pretty awesome rather than having more in-baskets on it.”(PCP)
	Physicians indicated preference for the in-basket format linked to a lipid panel report	
Content	Physicians highlighted the need for BPA and in-basket content to be more concise and clear	“There’s a lot of dense text. A lot of dense text…don’t be afraid of white space…and be telegraphic. So I would… critically looking at this message, I think I would look at individual words. This patient has an… those are not useful words yet. So LDL greater equal 190. That could be a line. Warning, possible familial hypercholesterolemia. Next line…Consider high-intensity… I wouldn’t even say high intensity… consider Rosuvastatin 20 or Atorvastatin 40 mg taken by mouth…Recommended laboratory testing could be another line.” (PCP)
	Only important information should be displayed	
	Have few clicks to access knowledge resources and a relevant order set	
	Have a reminder present to rule out secondary causes of hypercholesterolemia	
Timing & Frequency	Have ‘reasons not to use’ at the end of the CDS so that providers can explain their decision to not act upon the alert	“…but if I’m done with it [in-basket message] or I feel like I’ve addressed it, maybe I’ve put in the orders for the FH Clinic, I’d like to get it out of there, because otherwise things get too cluttered, and I’ll get very frustrated if there’s no way to dismiss it and if it just stays there forever.” (Specialist)
	Viewing one alert should turn off all other alerts for the same provider	
Prioritization	Most physicians described FH as being an important condition and agreed that color coding the CDS red would likely get their attention	“I think keep it red, because the people with this condition have significant events, and it’s something that you can prevent in their family members, so I would keep it red.”(Specialist)

**Table 4 jpm-10-00067-t004:** Implementation outcome measures by Weiner et al. and Consolidated Framework for Implementation Research (CFIR) constructs were assessed through a post-interview survey conducted with PCPs and specialists.

Measures and Constructs Assessed	Question	Completely Agree/Agree *N* (%)	Other ^1^ *N* (%)
Acceptability of Intervention Measure (AIM)	This tool meets my approval	11 (84.6)	2 (15.4)
	This tool is appealing to me	10 (76.9)	3 (23.1)
	I like this tool	11 (84.6)	2 (15.4)
	I welcome this tool	12 (92.3)	1 (7.7)
Intervention Appropriateness Measure (IAM)	This tool seems fitting	12 (92.3)	1 (7.7)
	This tool seems suitable	12 (92.3)	1 (7.7)
	This tool seems applicable	12 (92.3)	1 (7.7)
	This tool seems like a good match	10 (76.9)	3 (23.1)
Feasibility of Intervention Measure (FIM)	This tool seems implementable	11 (84.6)	2 (15.4)
	This tool seems possible	12 (92.3)	1 (7.7)
	This tool seems doable	12 (92.3)	1 (7.7)
	This tool seems easy to use	11 (84.6)	2 (15.4)
Intervention Characteristics	I trust the quality and validity of evidence supporting this intervention	12 (92.3)	1 (7.7)
	Implementing this tool is a good option to identify FH patients at Mayo	11 (84.6)	2 (15.4)
	This tool will improve early diagnosis of patients with FH	11 (84.6)	2 (15.4)
Outer Setting	This tool meets my needs to provide needed resources to my patients	9 (69.2)	4 (30.8)
Inner Setting	This tool is appropriate for ECH clinicians	11 (84.6)	2 (15.4)
	This tool fits within my existing workflow	10 (76.9)	3 (23.1)
	This tool will not increase the time needed with a patient	4 (30.8)	9 (69.2)
	The implementation of this intervention within Mayo is important	12 (92.3)	1 (7.7)
	I recognize the importance of implementing this tool into the practice	13 (100)	0 (0.0)
	This tool appears easy to access and incorporate into my workflow	10 (76.9)	3 (23.1)
Characteristics of Individuals	This is a valuable tool for ECH clinicians	10 (76.9)	3 (23.1)
	This tool will help me identify and refer or manage FH patients	12 (92.3)	1 (7.7)
Process	It is important to me that the cardiologists embedded in ECH continue to vet this tool	7 (53.8)	6 (46.2)

^1^ Other: (neither agree/disagree + completely disagree + disagree). Abbreviations: FH, Familial hypercholesterolemia; ECH, Employee and community health.

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
