# Peer review of "An Implementation Science Framework to Develop a Clinical Decision Support Tool for Familial Hypercholesterolemia"

_jpm, 2020, doi:10.3390/jpm10030067_

Round 1
Reviewer 1 Report
Thank you for the opportunity to review the article titled “An Implementation Science Framework to Develop a clinical Decision Support Tool for Familial Hypercholesterolemia.” This article covers an important issue and is will written.
I only have a few minor suggestions that I believe would improve the manuscript.
Were there any inclusion/exclusion criteria for the physician recruitment, other than working at the Mayo Clinic campuses?
Was the tool implemented in a primary care clinic, specialty clinic, or both?
It would be helpful to the readers to have an explanation of what best practice alert and in-basket message mean. Please provide examples of how these work?
Is there a certain specialty of physician that would typically test for FH? Also, is there a specific specialty of physicians the CDS would be implemented for?
Were the same physicians interviewed after each adjustment to the tool, or were new physicians interviewed each time?
The authors state that both the BPA and in-basket methods were implemented. Did both of them remain in use, or was one eliminated based on testing?
Author Response
We are appreciative of the comments and feedback provided by the Reviewer on our manuscript. We include our responses to the comments below. We have also uploaded the revised manuscript with the changes tracked. Please note that the line numbers listed below are concordant with the manuscript when viewed without markup.
Response to Reviewer
Thank you for the opportunity to review the article titled “An Implementation Science Framework to Develop a Clinical Decision Support Tool for Familial Hypercholesterolemia.” This article covers an important issue and is will written.
I only have a few minor suggestions that I believe would improve the manuscript.
(1) Were there any inclusion/exclusion criteria for the physician recruitment, other than working at the Mayo Clinic campuses?
Inclusion criteria for physician recruitment included those who were working at the Rochester, Minnesota campus of Mayo Clinic. We also limited recruitment to physicians from primary care, specifically the departments of family medicine and community internal medicine and from specialty care specifically cardiology and vascular medicine. Physicians from other Mayo Clinic campuses (Florida and Arizona) and other departments were not included from the study.
To clarify the physician inclusion criteria, the following changes have been made to Materials and Methods, subsection 2.1 Participants and recruitment on page 2 (line 85-89):
“To achieve the target sample size, we conducted purposive sampling with the following inclusion criteria: i) staff physicians at the Mayo Clinic campus in Rochester, Minnesota; ii) from primary care including the departments of family medicine and community internal medicine as well as specialty fields such as cardiology and vascular medicine; and iii) with previous experience using institutional CDS tools.”
(2) Was the tool implemented in a primary care clinic, specialty clinic, or both?
The CDS tool was implemented in ‘silent mode’ whereby the alert was set to not display for providers (it would display silently in the EHR background, inaccessible to providers) for the purpose of gathering initial triggering metrics. The algorithm that triggered the alerts was set to trigger only for providers in primary care (family medicine and community internal medicine), general internal medicine and cardiovascular medicine. Silent mode deployment of the CDS included all Mayo Clinic sites (Rochester, Arizona, Florida) and the Mayo Clinic Health System.
The following has been added to the Materials and Methods, subsection 2.4. Deployment of CDS in silent mode on page 3 (line 131-133):
“The silent mode setting would trigger the CDS alert in the EHR for providers in primary care, general internal medicine and cardiovascular medicine; however the alert was set to not display for providers, triggering only in the EHR background.”
(3) It would be helpful to the readers to have an explanation of what best practice alert and in-basket message mean. Please provide examples of how these work?
The following has been added to the Materials and Methods, subsection 2.2. Physician interviews and usability testing on page 2 and 3 (line 98-102):
“The first prototype was a best practice alert (BPA) – a clinical reminder with evidence-based management guidelines that would passively display to providers (requiring them to click on the alert to open it) at the point-of-care. The second prototype was an asynchronous in-basket alert – a type of inbox message that contained patient laboratory data and would be visible to providers upon logging into the EHR.”
(4) Is there a certain specialty of physician that would typically test for FH? Also, is there a specific specialty of physicians the CDS would be implemented for?
Providers in primary care are able to test for FH by ordering relevant laboratory tests as well as genetic testing. However, based on provider feedback during interviews, the majority indicated that were genetic testing required, they would prefer referral to a specialist. Specialists in cardiovascular medicine can also test for FH and patients can be referred to the FH clinic or lipid clinic to see a cardiologist with a specific focus on FH or lipidology.
When the CDS goes live across the institution, it will be deployed for all providers in primary care, general internal medicine and cardiovascular medicine. Those providers less likely to see patients with elevated cholesterol (e.g. rheumatologists, general surgeons) will be filtered out from receiving the alert to avoid alert fatigue and cognitive burden.
The following has been added to the Introduction on page 2 (line 52-54):
“Providers in both primary care as well as specialty clinics such as cardiology often perceive all cases of hypercholesterolemia similarly, failing to recognize the higher risk of coronary heart disease (CHD) and need for familial testing when the etiology is genetic.”
The following has been added to the Conclusion on page 11 (line 352-354):
“Employing an implementation science framework with qualitative interviews and usability testing, physician feedback was obtained to guide development of a CDS tool for FH and integrate it with the EHR for providers in primary care and cardiovascular medicine.”
(5) Were the same physicians interviewed after each adjustment to the tool, or were new physicians interviewed each time?
New physicians were interviewed after each round of CDS prototyping which enabled us to obtain a variety of perspectives from providers with differing clinical workflows as well as varied experiences in using CDS tools.
The following has been added to the Materials and Methods, subsection 2.2.Physician interviews and usability testing on page 3 (line 107-110):
“Four rounds of physician interviews and usability testing were conducted — new physicians, without prior exposure to the CDS tool, participated in each round and their feedback informed CDS prototyping and iterative refinements in subsequent rounds.”
(6) The authors state that both the BPA and in-basket methods were implemented. Did both of them remain in use, or was one eliminated based on testing?
Currently both BPA and in-basket alerts remain deployed in silent mode across all Mayo Clinic sites. Both alerts have different cues, the BPA appears when opening a patient chart, while the in-basket alert appears when a lipid panel is ordered and the patient meets ‘possible FH’ criteria (LDL-C ≥ 190 mg/dL in the absence of secondary causes of hypercholesterolemia). During the interviews, providers highlighted a need for both alert formats as this would increase the likelihood of them noticing the alert and acting upon it. We intend to test both alerts further in a randomized clinical trial, the results of which may impact whether both alerts remain implemented or not.
The following has been added to the Discussion on page 11 (line 334-338):
“A relatively large number of both BPA and in-basket alerts fired during silent mode deployment indicating a high burden of ‘possible FH’. This suggests a potentially large impact of CDS deployment on health outcomes. Currently both alerts remain deployed in silent mode and will be further evaluated in a clinical trial to assess the effect of CDS tools on outcomes such as changes in patient management post-deployment.”